# Therapeutic Potential of Highly Selective A_3_ Adenosine Receptor Ligands in the Central and Peripheral Nervous System

**DOI:** 10.3390/molecules27061890

**Published:** 2022-03-15

**Authors:** Elisabetta Coppi, Federica Cherchi, Martina Venturini, Elena Lucarini, Renato Corradetti, Lorenzo Di Cesare Mannelli, Carla Ghelardini, Felicita Pedata, Anna Maria Pugliese

**Affiliations:** Department of Neuroscience, Psychology, Drug Research and Child Health (NEUROFARBA), Division of Pharmacology and Toxicology, University of Florence, 50139 Florence, Italy; martina.venturini@unifi.it (M.V.); elena.lucarini@unifi.it (E.L.); renato.corradetti@unifi.it (R.C.); lorenzo.mannelli@unifi.it (L.D.C.M.); carla.ghelardini@unifi.it (C.G.); felicita.pedata@unifi.it (F.P.)

**Keywords:** adenosine receptors, oxygen and glucose deprivation, anoxic depolarization, neurotransmission, hippocampus, dorsal root ganglion neurons, visceral pain, N-type calcium channels, A_3_ receptor agonists, A_3_ receptor antagonists

## Abstract

Ligands of the G_i_ protein-coupled adenosine A_3_ receptor (A_3_R) are receiving increasing interest as attractive therapeutic tools for the treatment of a number of pathological conditions of the central and peripheral nervous systems (CNS and PNS, respectively). Their safe pharmacological profiles emerging from clinical trials on different pathologies (e.g., rheumatoid arthritis, psoriasis and fatty liver diseases) confer a realistic translational potential to these compounds, thus encouraging the investigation of highly selective agonists and antagonists of A_3_R. The present review summarizes information on the effect of latest-generation A_3_R ligands, not yet available in commerce, obtained by using different in vitro and in vivo models of various PNS- or CNS-related disorders. This review places particular focus on brain ischemia insults and colitis, where the prototypical A_3_R agonist, Cl-IB-MECA, and antagonist, MRS1523, have been used in research studies as reference compounds to explore the effects of latest-generation ligands on this receptor. The advantages and weaknesses of these compounds in terms of therapeutic potential are discussed.

## 1. Introduction: Adenosine and Its Receptors

Adenosine is a ubiquitous endogenous neuromodulator and is recognized as one of the most evolutionarily ancient and pervasive signaling molecules [1]. In particular, adenosine plays a crucial role in neuron-to-glia communication in both the central and peripheral nervous systems (CNS and PNS, respectively) [2,3,4] as well as in inflammatory processes throughout the periphery [5]. The effects of adenosine are mediated by the activation of four G protein-coupled receptors (GPCR): A_1_, A_2A_, A_2B_ and A_3_ receptors (A_1_Rs, A_2A_Rs, A_2B_Rs and A_3_Rs). A_1_Rs, A_2A_Rs and A_2B_Rs are fairly conserved throughout evolution, showing 80–95% homology among species, whereas A_3_Rs present high inter-species variability [6], with only 74% sequence homology between rat and human [7]. In general, A_1_Rs and A_3_Rs are coupled to adenylyl cyclase (AC) inhibition through G proteins of the Gi and Go family [8,9] (Figure 1), whereas A_2A_R and A_2B_R actions rely on the stimulation of adenylyl cyclase by Gs or G_olf_. However, beyond these canonical signaling mechanisms, the stimulation of A_1_Rs and A_3_Rs can also elicit Gq-mediated release of calcium ions from intracellular stores [10]. On the other hand, in addition to Gs, A_2B_R receptors can activate phospholipase C, which also occurs through Gq [10] (Figure 1). Moreover, all adenosine receptors are coupled to mitogen-activated protein kinase (MAPK) pathways, which include extracellular signal-regulated kinase 1 (ERK1), ERK2, JUN N-terminal kinase and p38 MAPK [10].

The activation of each adenosine receptor subtype depends upon extracellular adenosine levels, which, in turn, are regulated by a number of intra- or extracellular adenosine-synthetizing and degrading enzymes, as well as an uptake system at the membrane level. This complex machinery allows tissues to modulate purinergic signaling according to variations in tissue health status [11]. In the extracellular space, the ectonucleotidases CD39 and CD73 metabolize ATP and ADP to AMP, and AMP to adenosine, respectively, and they are the major sources of extracellular adenosine [12]. On the other hand, adenosine is removed by enzymes dedicated to its degradation, such as adenosine deaminase (ADA), which degrades extracellular adenosine to inosine (Figure 1). These catabolic enzymes, together with equilibrative nucleoside transporters (ENTs) and concentrative nucleoside transporters (CNTs) on the cell membrane, shunt extracellular adenosine into the intracellular space, thereby terminating adenosine receptor signaling [13,14].

Under physiological conditions, endogenous adenosine levels fall into the nanomolar range in the vast majority of organs and tissues [15]. However, during trauma or injury, particularly during hypoxic/ischemic insults [16,17,18], adenosine is released in massive quantities by damaged cells and reaches micromolar concentrations that can activate all subtypes of adenosine receptors [11].

### 1.1. Adenosine Receptors in Peripheral Tissues

In the periphery, adenosine receptors modulate a number of events, including inflammation, metabolism and cell-to-cell signaling. A pervasive action of peripheral adenosine is on A_1_Rs in the heart (Figure 2), where they are highly expressed and mediate potent bradycardia. For this reason, adenosine is administered as an emergency drug in arrhythmic conditions, e.g., paroxysmal supraventricular tachycardia [19].

It is also worth noting that adenosine is one of the most powerful endogenous anti-inflammatory agents thanks to its action on A_2A_Rs [5]. Indeed, A_2A_Rs are highly expressed in inflammatory cells, including lymphocytes, granulocytes and monocytes/macrophages, where they reduce the release of pro-inflammatory cytokines, e.g., tumor necrosis factor-alpha (TNFα), interleukin-1β (IL-1β), IL-6 [20] and IL-12 [21], and enhance the release of anti-inflammatory mediators, such as IL-10 [22]. Furthermore, A_2A_R activation on blood vessels is a powerful hypotensive stimulus due to its vasodilating action via intracellular cAMP increase [23] (Figure 2). 

Concerning the A_2B_R subtype, the main A_2B_R-mediated effect in the periphery is known to be in the airways, where this receptor is highly expressed and mediates robust bronchoconstriction [24]. Indeed, methylxanthine theophylline, a non-selective adenosine receptor antagonist, is a second-line bronchodilator in asthma therapy [25] (Figure 2).

A_3_R is the most variable adenosine receptor subtype among mammalian species in terms of pharmacology and tissue distribution [6,26]. As an example, rat A_3_R is resistant to blockade by xanthines, the typical adenosine receptor antagonist class, but human and sheep A_3_Rs can be potently blocked by these compounds [6]. Indeed, important differences in tissue distribution among species are also reported for this adenosine receptor subtype, as rat and human dorsal root ganglion (DRG) neurons express high levels of A_3_Rs, whereas mouse DRGs are devoid of them [27,28]. The distribution of A_3_Rs throughout the body is sparse and discontinuous. High levels of A_3_R expression are found in the testis, uterus and spleen [29]. Relatively low levels of A_3_R are present in the heart, brain, neurons, lung and colon [29]. One of the first documented actions of this adenosine receptor subtype is its activation of mast cell degranulation. Indeed, A_3_Rs are highly expressed in rodent mast cells, where they represent one of the most efficacious stimuli for histamine release [30]. However, due to species-specific differences, A_3_R activation does not induce degranulation and histamine release in human mast cells [31] (Figure 2). Finally, abundant A_3_R transcripts have been found in livers from humans but not from other species, and moderate levels are found in the heart, kidney and placenta in rodents [6].

### 1.2. Adenosine Receptors in the CNS

At the central level, A_1_Rs are the primary effectors of adenosine actions in the CNS, where they are widely distributed, with the highest levels reported in the cerebral cortex and hippocampus [29]. A_1_Rs are known to inhibit neurotransmitter release by inhibiting presynaptic voltage-gated calcium channels (VGCCs) [32,33] and to decrease neuronal excitability by increasing postsynaptic potassium conductance [34]. During brain ischemia, these actions provide important adenosine-mediated neuroprotection by maintaining cells in a low energy consumption state and by counteracting excessive glutamate release, which is responsible for the excitotoxic damage [35].

The functional effect of the Gs-coupled A_2A_R subtype in the CNS is the opposite to that of A_1_Rs, as these receptors are reported to enhance glutamate release by facilitating calcium entry through presynaptic VGCCs and inhibiting its uptake [36]. Moreover, A_2A_R inhibits voltage-dependent potassium channels, thus promoting cell excitability and neurotransmitter release [37] and exacerbating excitotoxic damage during ischemic/hypoxic conditions [37]. The expression levels of A_2A_R in the CNS are comparable to those of A_1_Rs only in the caudate/putamen nuclei, where they play a deleterious role in the acute phases of an ischemic insult by increasing brain damage and neurological deficit [38] after middle cerebral artery occlusion (MCAo) in rats [16].

The cognate A_2B_R subtype presents similar features to A_2A_Rs: it is coupled to Gs proteins and promotes glutamate release in the brain, for example, in the CA1 hippocampus [39], where it has been identified at presynaptic sites [36,39]. Furthermore, A_2B_Rs are expressed at remarkable levels by neurons and glia in the thalamus, lateral ventricles and striatum [40]. However, due to its low (micromolar) affinity for the endogenous agonist, the contribution of this receptor subtype might become evident only in conditions of tissue damage or trauma, when high amounts of adenosine are released [41]. The facilitatory effect of A_2B_Rs on glutamatergic neurotransmission was assessed in acute hippocampal slices by using the electrophysiological protocol of paired-pulse facilitation (PPF), which is known to modulate short-term synaptic plasticity. Our research group recently confirmed that A_2B_Rs decreases PPF, thus enhancing glutamate release, in an A_1_R-dependent manner. Indeed, the effect of BAY60-6583 was prevented not only by the A_2B_R antagonists MRS1754 and PSB-603 but also by the A_1_R blocker DPCPX [42]. Furthermore, we extended the results to a newly synthetized BAY60-6583 analog, the A_2B_R-selective agonist P453, recently described in [43], which proved to have higher affinity than BAY60-6583 [42]. In line with the above data, A_2B_R blockade proved effective in preventing oxygen and glucose deprivation (OGD)-induced neurotoxicity in CA1 rat hippocampal slices [44] and improved oligodendroglial maturation in an in vitro model of rat oligodendrocyte progenitor cell (OPC) cultures [45].

A_3_Rs in the CNS have gained attention in the last decades for their encouraging therapeutic potential. As an example, overexpression in cancer and inflammatory cells has led to A_3_R being proposed as a tumor marker or as a target for personalized anti-cancer treatment programs [46]. The A_3_R subtype also plays important roles in chronic pain reduction, neuroprotection and several cardiovascular processes and has been associated with neurological, cardiovascular, inflammatory and autoimmune diseases. Importantly, A_3_R agonists are in clinical trials for rheumatoid arthritis, plaque psoriasis, hepatocellular carcinoma, non-alcoholic fatty liver disease and non-alcoholic steatohepatitis [47,48] and have already shown a safe and secure pharmacological profile [49], with no apparent invasive side effects [50]. The fact that A_3_R agonists proved to be devoid of significant cardiovascular side effects in clinical trials so far renders this adenosine receptor subtype an eligible target to obtain adenosine-mediated protective effects, for example, in ischemia or chronic pain, without undergoing A_1_R-mediated bradycardia or hypotension nor A_2A_-mediated vasodilation [34]. Furthermore, conflicting results obtained with A_2_R-selective agonists in preclinical models discouraged the attempt to develop a clinical approach with these compounds [51].

## 2. Therapeutic Potential of A_3_R Ligands

### 2.1. A_3_R Ligands as Therapeutic Tools in Brain Ischemia

There are no current treatments for stroke that are able to prevent or alleviate neuronal loss and tissue damage in the acute or post-ischemic phases after the insult. The only therapy strategies currently in use are the administration of tissue plasminogen activator (tPA) within the first 3–4 h after the insult [52] and, in some cases, hypothermia [53,54]. As stated above, adenosine is released in great amounts during hypoxic/ischemic conditions and plays a pervasive role in this pathology by activating all subtypes of P1 receptors. Both protective and deleterious effects have been described upon their selective activation, depending on the receptor subtype and timing after the insult.

The most widely recognized effect of adenosine during brain ischemia is the neuroprotective role exerted by the Gi-coupled A_1_R subtype, which is known to inhibit exaggerated glutamate release during excitotoxic damage in the ischemic core and penumbra [33,34]. As a result of the wide distribution of A_1_Rs within the brain, they have a higher impact compared to A_2A_R, A_2B_R and A_3_R subtypes [18]. Unfortunately, due to unacceptable side effects at the peripheral level (bradycardia and hypotension), A_1_R-selective agonists are not suitable for clinical use in preventing post-ischemic damage [34].

Hence, research was driven towards the other Gi-coupled adenosine receptor subtype, A_3_R. Interestingly, A_3_R has been proposed as a novel therapeutic target for a number of pathologies [42,43], and A_3_R-selective nucleosides are already in phase 2 and/or 3 clinical trials for autoimmune inflammatory diseases, liver cancer and non-alcoholic steatohepatitis (see www.clinicaltrials.gov; NCT00556894; NCT02927314; 12 March 2022) [49,50]. Notably, activation of A3R in humans by the selective and orally bioavailable A3R agonist IB-MECA (1-deoxy-1-[6-[[(3-iodophenyl)methyl]amino]-9H-purine-9-yl]-N-methyl-β-D-ribofuranuronamide) and its chlorinated counterpart Cl-IB-MECA (2-chloro-N6-(3-iodobenzyl)-adenosine-5-N-methyluronamide) is not associated with cardiac or hemodynamic effects [50], and they have shown encouragingly safe profiles [49]. Further attention was focused on this adenosine receptor subtype for the investigation of innovative pain-relieving strategies, as detailed below, due to evidence for their good “safety and security” profile.

Given their clinical potential, significant effort has been invested into identifying potent A_3_R ligands with high subtype selectivity and minimal species variability [55,56]. Currently, two A_3_R agonists, N6-(3-iodobenzyl)adenosine-59-N-methyluronamide (IB-MECA) and 2-chloro-N6-(3-iodobenzyl)-adenosine-59-N-methyluronamide (Cl-IB-MECA), are in clinical trials for the treatment of psoriasis, rheumatoid arthritis, dry-eye syndrome and hepatocellular carcinoma [42]. On this encouraging clinical basis, recent knowledge about their role in cerebral ischemia prompted research on new strategies based on A_3_R-selective ligands in the treatment of stroke. From a molecular point of view, the structure–activity relationship of A3R agonists has been investigated extensively [56]. It has been reported that the affinity and selectivity of A3R agonists can be enhanced through the substitution of the ribose tetrahydrofuryl group with a rigid bicyclo[3.1.0]hexane (methanocarba) ring system, the addition of m-substituted benzyl groups at the N6 position or the addition of alkyn-2-yl groups to the C2 position [57,58,59,60]. The North (N)-methanocarba ring system retains the conformation of the ribose ring that is favoured at A_3_R. Furthermore, the subtype selectivity of A_3_R agonists can be enhanced through 59-N-methyluronamide substitution [61]. Collectively, these studies provide a valuable framework, particularly with respect to the N6 and C2 positions, for the rational design of potent A_3_R agonists with high efficacy and subtype selectivity.

Concerning brain ischemia, basic research has produced controversial data about the role of A_3_Rs in this pathology. Early studies pointed to a protective effect of this receptor subtype, as A_3_R^−/−^ transgenic mice showed increased neurodegeneration in response to hypoxia [62]. Consistent with this report, [63] demonstrated that the selective activation of A_3_Rs in rat cortical neurons is involved in the inhibition of excitatory neurotransmission, suggesting that this receptor subtype contributes to the well-known adenosine-mediated neuroprotection during ischemia brought about by the A1R subtype. At variance with this result, further evidence showed that acute A_3_R stimulation exacerbates in vivo ischemic damage [64], indicating a deleterious role of this receptor subtype during the insult. Similarly, in vitro models of brain ischemia obtained by oxygen and glucose deprivation (OGD) demonstrated that the blockade of A_3_Rs prevents synaptic failure and excitotoxic damage induced by the insult and detected as anoxic depolarization (AD) [65] (for a review, see: [66]) in the CA1 hippocampus [67]. This concept is in line with a previous observation demonstrating that A_3_R activation during OGD limits the beneficial effects of ischemic preconditioning on the resistance of synaptic transmission to ischemia-like insults in hippocampal slices [67]. The neuroprotective effect observed by A_3_R blockade during OGD was exerted not only by the prototypical A_3_R antagonist MRS1523, a pyridine derivative [67], but also by the A_3_R antagonist LJ1251, a highly selective nucleoside derivative synthetized by Kenneth Jacobson and his team at the National Institutes of Health (NIH; Bethesda, MD, USA) [68].

The situation is further complicated by the fact that A_3_Rs not only activate Gi proteins but, under certain conditions, also activate Gq proteins, according to a previous report [5], which leads to an increase in intracellular calcium ions, triggering deleterious intracellular processes.

The contradictory role of A_3_Rs during hypoxic/ischemic insults reported above is in line with an equally controversial effect of A_3_R activation in the brain under normoxic conditions. Indeed, selective A_3_R agonists have been reported to induce either inhibitory [69] or facilitatory effects on glutamatergic neurotransmission [70,71,72,73,74].

In an attempt to summarize the above-mentioned data, we can conclude that A_3_R stimulation during prolonged OGD is protective in the first phases of the insult by participating, together with A_1_Rs, in the (Gi-mediated) inhibition of excitatory neurotransmission [68], whereas, at later times, prolonged receptor activation might lead to (Gq-mediated) intracellular calcium accumulation, contributing to cell damage (for a review, see: [16]). In this view, even if agonists of A_3_Rs are neuroprotective in the acute phases of stroke, at later times, corresponding to when most ischemic patients approach clinics, antagonists are the most advantageous strategy for an eventual post-ischemic A_3_R-based therapy because they might protect neurons, especially those “salvageable” in the penumbral area, from neurodegeneration and death [75,76,77].

Finally, it is worth noting that recent findings from basic research highlighted a promising neuroprotective effect exerted by a new compound, a mixed and partial agonist of A_1_Rs and A_3_Rs, known as AST-004 (MRS4322), a low-molecular-weight nucleoside metabolite recently synthetized by the group of Kenneth Jacobson [78]. Experimental models of stroke performed in mice [78] or non-human primates [79] demonstrated that treatment with the A_1_R/A_3_R agonist after acute ischemic stroke resulted in significantly reduced lesion volume, without alterations in cardiovascular parameters. The cerebroprotective effect of the compound was also found to correlate with unbound AST-004 concentrations in the plasma and cerebrospinal fluid, as well as estimated brain A_1_R and A_3_R occupancy, indicating the activation of adenosine A_1_Rs and/or A_3_Rs [79]. Such encouraging data prompted the evaluation of AST-004 in a Phase I clinical trial for stroke (https://www.astrocytepharma.com/2021/11/astrocyte-announces-publication-in-journal-stroke/; 12 March 2022).

### 2.2. A_3_R Ligands as Therapeutic Tools in Chronic Pain

Chronic pain is a highly debilitating condition, disturbing all aspects of our daily experience, from social life to career-related contexts. The pharmacological tools available to date are sometimes inadequate or, as in the case of opioids, limited by serious adverse effects [80]. In an effort to find innovative, non-opioid pain-relieving compounds, many experimental reports have identified adenosine receptors as potential targets for acute or chronic pain management.

The first proof of adenosine’s anti-nociceptive effect dates from the 1970s, when administration of adenosinergic agonists proved effective in pain control. These studies emphasized the role of adenosine A_1_Rs in producing anti-nociceptive effects, with some effects ascribed to the A_2A_R subtype [81,82]. Adenosine involvement in peripheral nociception was further confirmed; e.g., the local administration of exogenous A_1_R agonists to the hind paw of the rat produces anti-nociceptive effects in a pressure hyperalgesia model [83], whereas local administration of A_2_R agonists enhances pain responses [84], an action due to adenosine A_2A_R activation, as confirmed by using the selective agonist CGS21680 [85]. Later, it was demonstrated that the anti-nociceptive action of A_1_R agonists could be ascribed to AC inhibition and to the consequent decrease in cAMP production in sensory nerve terminals [86,87]; thus, a robust protective role of A_1_R agonists emerged [88]. On the other hand, the A_2A_R-mediated promotion of cutaneous pain resulted from the stimulation of AC, leading to increased cAMP levels in sensory nerve terminals [86,87], thus producing opposite effects to those elicited by the anti-hyperalgesic, Gi-coupled A_1_R subtype. However, the relation between A_2A_Rs and pain has been controversial, with evidence supporting either pro-nociceptive or anti-nociceptive activity depending on the receptor localization and animal models of pain [88]. Indeed, a relevant A_2A_R-mediated anti-nociceptive effect was described in a recent study demonstrating that central neuropathic pain evoked by dorsal root avulsion could be reversed by a single intrathecal injection of A_2A_R agonists [89]. The beneficial effects of A_2A_R agonists in this particular experimental model were associated with reduced reactive gliosis in the CNS. At variance with this finding, A_2A_R antagonists reduced chemotherapy-induced neuropathic pain when administered orally [90]. The discrepancies between the reported effects of A_2A_Rs in pain control could be due to the possible opposing roles that this adenosine receptor subtype exerts in the periphery (anti-inflammatory effect) versus the CNS (pro-excitatory effect) (for reviews, see [91]). On this basis, the idea took shape that AC-stimulating receptors were pro-algesic factors, whereas AC-inhibiting messengers would act as anti-nociceptive signals. In line with this assumption, A_2B_R activation promotes pain states by increasing the release of interleukin-6 (IL-6) [92,93], a pro-inflammatory cytokine also known to cause nociceptor hyperexcitability [92]. In this regard, it should be noted that adenosine-mediated signaling in inflammation and pain control has been recently linked to the nuclear orphan receptor 4A (NR4A)-dependent pathway [94]. Indeed, it has been demonstrated that A_2A_R activation counteracts NR4A2 and NR4A3 gene induction in a human mast cell line, whereas, after A_2A_R silencing or in the presence of the A_2A_R antagonist SCH58261, the adenosine analog NECA amplifies NR4As induction, thus suggesting that A_2A_R activation counteracts NR4A2 and NR4A3 induction in mast cells, whereas the activation of other AR subtypes (i.e., A_2B_Rs and/or A_3_Rs) induces the upregulation of these factors [94]. Such data suggest that the role of adenosine in mast-cell-related inflammatory events may be linked to differential nuclear orphan receptor 4A axis modulation, depending on the adenosine receptor subtype stimulated. Another piece of evidence about the role of adenosine in inflammation and pain came from recent data showing that the CCI-induced increase in iNOS, bax/bcl2, iba-1 and TNF-α expression in the lumbar spinal cord of rodents was attenuated by allopurinol, an effect reversed by the A_1_R-selective and the unselective adenosine receptor antagonist theophylline and by the A_1_R-selective blocker DPCPX. Of note, the mechanical anti-allodynic effect of allopurinol was only prevented by theophylline, indicating, even in this experimental mode, a differential role of distinct adenosine receptor subtypes [95].

Another possible pathway of adenosine-mediated pain control has been proposed to consist in direct TRPV1 inhibition by directly interacting with the receptor protein [96]. In particular, it was demonstrated that capsaicin-induced inward currents in DRG neurons were inhibited by adenosine and that A_1_Rs on DRG neurons colocalized with TRPV1 as a membrane microdomain in allodynic mice, with the level of colocalization correlating with the development of the symptom [97].

Of note, mechanical muscle hyperalgesia is a relatively common debilitating condition in which the classical inflammatory response (release of bradykinin, prostaglandins, pro-inflammatory cytokines, etc.) plays a major role. In this regard, it has been demonstrated that A_2B_R stimulation in rat skeletal muscle cells induces IL-6 release via a cAMP-dependent pathway, thus indicating A_2B_R antagonists as potentially important pharmacological targets in treating inflammation and related diseases in skeletal muscle tissues [93,98].

However, from a clinical point of view, the translation of A_1_R- or A_2_R-selective ligands for pain relief therapy is unfortunately hindered by important side effects due to the expression of A_1_Rs at the cardiac level or A_2A_Rs in vascular smooth muscle, e.g., bradycardia and vasodilation, respectively [34]. Furthermore, data on the A_2A_R-mediated role in pain were elusive and often contradictory [51], thus failing to materialize into a clinical approach. Notably, agonists of A_3_R have shown important pain-relieving properties in preclinical settings of several models of chronic pain, as detailed below.

In a mouse model of neuropathic pain, it was demonstrated that an intraperitoneal (i.p.) injection of Cl-IB-MECA in mice or rats on day 7 after injury proved effective in relieving neuropathic pain caused by chronic constriction injury (CCI) or chemotherapy treatment in rodents [99]. These results were confirmed by using a number of highly selective, in some cases water-soluble, A_3_R agonists recently synthesized by the group of Kenneth Jacobson [100], among which are the “first in class” compounds MRS5980 [101] and MRS5698 [102,103] and the water-soluble prodrug MRS7476 [104].

The pain-reliving effects of A_3_R agonists described above were sensitive to the A_3_R antagonist MRS1523 but not to the A_1_R and A_2A_R blockers DPCPX and SCH-442416, respectively, nor to the opioid antagonist naloxone [99], and were absent in A_3_R^−/−^ mice [105]. It is also worth noting that A_3_R-mediated pain-relieving effects present a certain degree of sexual dimorphism in rats [106] and do not modify the physiological pain threshold under control conditions, indicating a specific anti-hyperalgesic effect in conditions of altered sensitivity, with no analgesic effects per se [99,100,101,102,103,104,105].

Both central and peripheral mechanisms of action have been proposed in order to explain the pain-relieving properties of A_3_R stimulation. Recent evidence [107] demonstrated that selective A_3_R agonism (by Cl-IB-MECA or MRS5980) in isolated rat DRG neurons decreased neuronal firing and inhibited pro-nociceptive calcium currents sensitive to the N-type channel blocker PD732121, an analog of ω-conotoxin (ω-CTX) [107] (Figure 3). In the same work, it was demonstrated that the selective activation of A_1_Rs by the selective agonist N6–cyclopentyladenosine (CPA) still produced calcium current (I_Ca_) inhibition, but the extent of this effect was by far smaller than that achieved by the A_3_R agonist Cl-IB-MECA, even at maximum concentration. In addition, the inhibitory effect of adenosine on I_Ca_ was blocked to a greater extent by the A_3_R-selective antagonist VUF5574 than by the A_1_R-selective blocker 8-cyclopentyl-1,3-dipropylxanthine (DPCPX) [107], thus demonstrating that either A_1_R or A_3_R activation might inhibit N-type I_Ca_ (Cav_2.2_) in rat DRG neurons, but, when the endogenous agonist adenosine is released, the A_3_R-mediated response prevails. These results are in agreement with data from the 1980s showing that adenosine inhibits voltage-dependent calcium channels (VDCCs) in isolated mouse DRG neurons [108], an effect that was only partially ascribed to A_1_R activation [109]. Hence, by using the in vitro DRG model, it was demonstrated that A_3_R activation inhibits calcium entry into the neurons and action potential firing, suggesting the inhibition of synaptic transmission in the dorsal horns. These results are also in line with previous evidence demonstrating that aberrant expression and/or activity of N-type VDCCs is associated with neuropathic pain [109] and with the fact that ziconotide, a derivative of ω-CTX, was FDA-approved in 2000 (Prialt) for intrathecal treatment of severe and refractory chronic pain [110,111]. However, severe adverse effects (e.g., hallucinations or other psychiatric symptoms) are associated with a direct calcium channel block, likely due to their wide expression in neurons throughout the CNS.

The efficacy of A_3_R stimulation was also recently demonstrated in a colitis-induced model of visceral pain, and consistent with the above data, a similar mechanism of action was proposed. The authors showed that, when experimental colitis was reproduced in rats by intracolonic instillation of dinitrobenzenesulfonic acid (DNBS), a single injection of Cl-IB-MECA, or the more selective A_3_R agonist MRS5980, at the peak of visceral hypersensitivity (day 14 after DNBS injection) was effective in relieving visceral allodynia, measured by quantifying the number of abdominal muscle contractions upon intestine distention. Neither compound was found to affect gastrointestinal transit. Interestingly, the effect of A_3_R activation was mimicked not only by the clinically used drug linaclotide, a compound currently used to control abdominal pain in irritable bowel syndrome patients [112,113], but also by the N-type VDCC blocker PD732121 [114], thus pointing to a role of these calcium channels in A_3_R-mediated visceral pain relief as well.

A_3_R agonists were further shown to prevent the development of chemotherapy-induced neuropathic pain (CINP) by exerting beneficial effects associated with the modulation of spinal neuro-glial communication and neuroinflammatory processes. The A_3_R agonist IB-MECA attenuated paclitaxel-induced neuropathic pain in mice by decreasing astrocytic activation in the spinal cord [115] as a result of decreased levels of neuroexcitatory/pro-inflammatory cytokines such as TNF-α and interleukin-1β (IL-1β), whereas an increase in the neuroprotective/anti-inflammatory interleukin-10 (IL-10) release was observed [115] (Figure 3). Similarly, microglial activation by oxaliplatin-induced mechano-hypersensitivity in rats was not observed in a rat model of CINP [116]. These data suggest that inhibition of glial-associated neuroinflammation in the spinal cord contributes to the protective actions of A_3_R.

Concerning peripheral effects of A_3_Rs in pain control, a different peripheral pathway of A_3_R-mediated anti-hyperalgesia has been recently identified. It was found that transgenic Rag^−/−^ mice, lacking T and B cells, are insensitive to the anti-allodynic effects of the A_3_R agonist MRS5980, whereas adoptive transfer of CD4^+^ T cells from wild-type (wt) mice infiltrated the inflamed DRG and restored A_3_R agonist-mediated anti-allodynia [116]. Of note, adoptive transfer of CD4^+^ T cells from A_3_R^−/−^ or IL-10^−/−^ mice did not restore the A_3_R-mediated effect, demonstrating that A_3_R activation in CD4^+^ T cells elicits IL-10 release, which, in turn, is responsible for the anti-hyperalgesic effect of the A3R agonist MRS5980 [117]. Further downstream mechanisms were elucidated in the same work by the use of an in vitro model consisting of co-cultures of DRG neurons plus T cells isolated from mice, either naïve or CCI animals. With this model, CD4^+^ or CD8^+^ T-cell infiltration into the ganglion was reproduced in vitro by a circumscribed experimental model in which any final effect of cytokine(s) released by T cells on DRG neurons upon A_3_R stimulation could be evaluated in isolation. Notably, application of the A_3_R agonist MRS5980 to the co-culture significantly reduced the action potential (AP) firing evoked by a depolarizing current ramp in neurons. The effect was not observed in DRG-CD8^+^ T-cell co-cultures, nor in DRG neurons cultured alone, demonstrating the selective engagement of A3Rs expressed on CD4^+^ T cells [117]. Importantly, the effect of MRS5980 on neuronal firing was also abolished by pre-treatment with an anti-IL-10-selective antibody, thus unequivocally pointing to this anti-inflammatory cytokine as the main effector produced by CD4^+^ T cells upon A_3_R activation to inhibit neuronal excitability (Figure 3) [117]. Interestingly, the involvement of IL-10 in A_3_R-mediated pain control was already evident in a previous work, where attenuating IL-10 signaling in oxaliplatin-treated rats was achieved with an intrathecal neutralizing IL-10 antibody or in IL-10^−/−^ mice [103].

Hence, it appears that the modulation of A_3_Rs induces potent anti-hypersensitive effects in diverse preclinical models of chronic pain. Nevertheless, the mechanisms by which this receptor subtype exerts anti-hyperalgesic and anti-allodynic effects remain to be clarified, and, thus far, both peripheral and central effects have been described, with some gender-specific differences in rats. Of note, the efficacy of A_3_R ligands for chronic pain control is particularly relevant, as tolerance and significant side effects do not occur.

**Figure 3 molecules-27-01890-f003:**
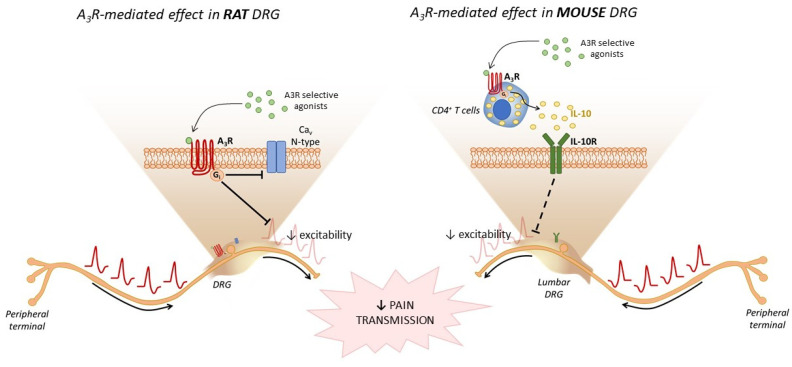
Adenosine A_3_ receptors and pain control. A_3_ receptors (A_3_Rs) are expressed on rat DRG neurons, and their activation by the selective agonist MRS5980 decreases action potential (AP) neuronal firing and inhibits N-type voltage-gated calcium channels [110]. A_3_Rs expressed on CD4^+^ T cells, but not on mouse DRG neurons, promote interleukin-10 (IL-10) release that, by activating IL-10 receptors (IL-10R) on DRG neurons, reduces neuronal excitability by inhibiting AP firing [117].

## 3. Conclusions

In summary, we can conclude that A_3_Rs are emerging as promising targets for the treatment of a number of pathologies due to the limited side effects of their ligands in comparison to those targeting other adenosine receptor subtypes (i.e., A_1_Rs or A_2A_Rs). In particular, A_3_R antagonists proved effective in preclinical animal models of brain ischemia and OGD in hippocampal slices, thus paving the way for the development of new, highly selective A_3_R blockers for the treatment of stroke. Furthermore, valuable evidence from rodent models of chronic pain indicates the possible use of selective A_3_R agonists as non-narcotic anti-hyperalgesic agents for pain control.

## Figures and Tables

**Figure 1 molecules-27-01890-f001:**
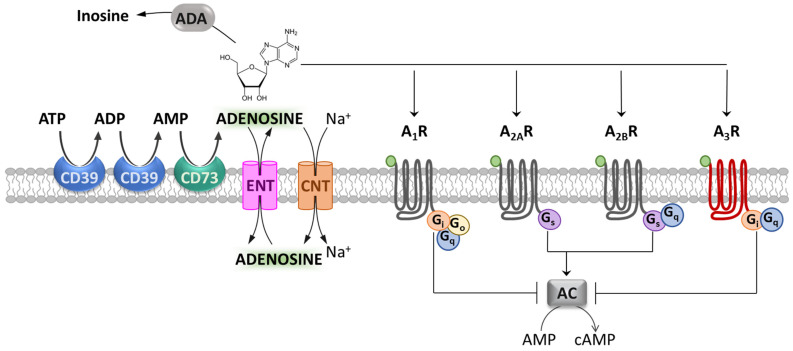
Adenosine and its receptors. Schematic representation of adenosine metabolism, transport and metabotropic receptors. The ectonucleotidases CD39 and CD73 metabolize ATP and ADP to AMP, and AMP to adenosine. The equilibrative nucleoside transporter (ENT) or concentrative nucleoside transporter (CNT) families mediate adenosine reuptake. Adenosine metabotropic receptors (A_1_, A_2A_, A_2B_ and A_3_ receptors: A_1_R, A_2A_R, A_2B_R and A_3_R) are differently coupled to adenylyl cyclase (AC) inhibition or stimulation. Adenosine deaminase (ADA) deactivates extracellular adenosine by converting it into inosine.

**Figure 2 molecules-27-01890-f002:**
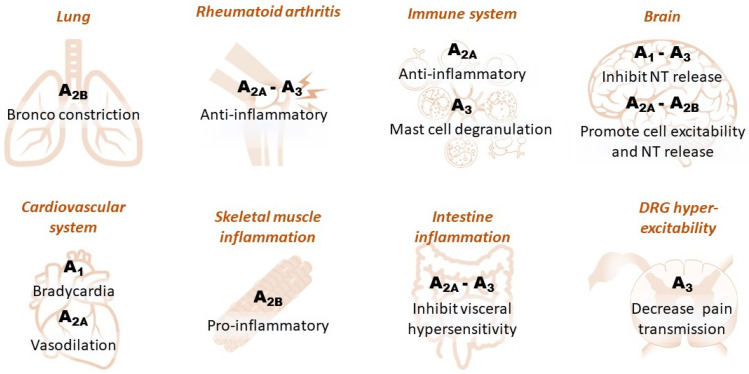
Schematic diagram illustrating the effect of different adenosine receptor subtypes in peripheral and central tissues. Dorsal root ganglia: DRG.

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
