# Peer review of "Therapeutic Potential of Highly Selective A_3_ Adenosine Receptor Ligands in the Central and Peripheral Nervous System"

_molecules, 2022, doi:10.3390/molecules27061890_

Round 1
Reviewer 1 Report
This is a knowledgeable and useful review of the therapeutic potential of selective A3 receptor ligands. The review particularly focussed on the potential roles of A3Rs in brain ischemia and chronic pain, and represented an up-to-date overview of the literature. The two figures were a good addition.
I have only a few comments to make:
As part of the introduction, I think there needs to be a bit more detail added into the species differences in A3-receptor pharmacology and the way that A3 receptors are expressed in different cells in different species. e.g. Figure 2 shows marked differences between A3 receptor expression and action between rat and mouse DRGs. The species differences in A3-receptor pharmacology (including human) should also be detailed more effectively. Perhaps include a table of affinities?
On page 2 line 81, there is a comment about the sparse distribution of A2B receptors in brain. I am not sure this is entirely true, and may depend on species. According to protienatlas.org, there is a reasonably widespread distribution of A2B receptors in human brain, which is usually higher than A2A receptors. Some further comment on this is required in the text.
On page 5 CCI needs defining (presumably chronic constriction injury?).
There is the odd reference to (ref) which needs addressing and some misspelt words need correcting.
Finally, I think the manuscript would benefit from a summary paragraph at the end.
Author Response
Reviewer 1
This is a knowledgeable and useful review of the therapeutic potential of selective A3 receptor ligands. The review particularly focussed on the potential roles of A3Rs in brain ischemia and chronic pain, and represented an up-to-date overview of the literature. The two figures were a good addition.
I have only a few comments to make:
As part of the introduction, I think there needs to be a bit more detail added into the species differences in A3-receptor pharmacology and the way that A3 receptors are expressed in different cells in different species. e.g. Figure 2 shows marked differences between A3 receptor expression and action between rat and mouse DRGs. The species differences in A3-receptor pharmacology (including human) should also be detailed more effectively. Perhaps include a table of affinities?
Our response. We thank the Reviewer for his/her positive comments on our paper and for rising important points that helped us to improve the manuscript. We agree that a more detailed description of A3R differences among species in terms of pharmacology and tissue distribution is needed in the Introduction. Concerning the suggested table of affinities of various A3R ligands in different species, we wouldn’t have valuable information to add to those tables already found in the literature (i.e.: Stefania Gessi, Stefania Merighi, Katia Varani, Edward Leung, Stephen Mac Lennan, Pier Andrea Borea. The A3 adenosine receptor: an enigmatic player in cell biology. Review Pharmacol Ther 2008 Jan;117(1):123-40. doi: 10.1016/j.pharmthera.2007.09.002. Epub 2007 Sep 22; Linden J. Cloned adenosine A3 receptors: pharmacological properties, species differences and receptor functions. Trends Pharmacol Sci. 1994 Aug;15(8):298-306. doi: 10.1016/0165-6147(94)90011-6; both of which are now quoted in the manuscript), so we decided to decline the Reviewer’s proposal.
So, as suggested, we added a new paragraph, as follows (page 3, lines 84-100):
“The A3R is the most variable adenosine receptor subtype among mammalian species in terms of pharmacology and tissue distribution. As an example, the rat A3R is resistant to blockade by xanthines, the typical adenosine receptor antagonist class, but human and sheep A3Rs can be potently blocked by these compounds [6]. Indeed, important differences in tissue distribution among species is also reported for this adenosine receptor subtype, as rat and human dorsal root ganglion (DRG) neurons express high levels of A3Rs where-as mouse DRG are devoid of them [27-28]. The distribution of A3Rs throughout the body is sparse and discontinuous. High levels of A3R expression are found in the testis, uterus and spleen [29]. Relatively lower levels of A3R are also present in the heart, brain, neurons, lung and colon [29]. One of the first documented actions for this adenosine receptor sub-type was its activation of mast cell degranulation. Indeed, A3Rs are highly expressed in rodent mast cells where they represent one of the most efficacious stimuli for histamine release [30]. However, still due to species-specific differences, A3R activation does not in-duce degranulation and histamine release in human mast cells [31]. Finally, Abundant A3R transcripts have been found in liver from humans, but not from other species, and moderate levels are found in heart, kidney and placenta in rodents [6].”
On page 2 line 81, there is a comment about the sparse distribution of A2B receptors in brain. I am not sure this is entirely true, and may depend on species. According to protienatlas.org, there is a reasonably widespread distribution of A2B receptors in human brain, which is usually higher than A2A receptors. Some further comment on this is required in the text.
Our response. We thank the Reviewer for alerting us about this misunderstanding on A2BR distribution. After a more detailed analysis of the present literature, we modified our sentence by deleting the comment on the scarce A2BR distribution in the brain. Indeed, as emerged from recent literature, we added the following sentence (page 3 lines 121-123):
“Furthermore, A2BRs are expressed at remarkably levels by neurons and glia in the thalamus, lateral ventricles and striatum [43]”.
On page 5 CCI needs defining (presumably chronic constriction injury?).
Our response. We thank the Reviewer to underline this mistake. We added the definition of the acronym “CCI”, as follows: “chronic constriction injury (CCI)” (page 7 line 335).
There is the odd reference to (ref) which needs addressing and some misspelt words need correcting.
Our response. We thank the Reviewer to underline this mistake. We added reference n. 118 in the missing point (page 6 line 273).
Finally, I think the manuscript would benefit from a summary paragraph at the end.
Our response. As suggested, we added a summary paragraph, as follows (page 9 lines 426-435)
“3. Conclusions.
In summary, we can conclude that A3Rs are emerging as promising targets for the treatment of a number of pathologies due to scares side effects of their ligands in comparison to those targeting other adenosine receptor subtypes (i.e. A1Rs or A2ARs). In particular, A3R antagonists proved effective in preclinical animal models of brain ischemia or OGD in hippocampal slices, thus paving the way for the development of new, highly selective, A3R blockers for the treatment of stroke. Furthermore, valuable evidence from rodent models of chronic pain indicate the possible use of selective A3R agonists as non-narcotic anti-hyperalgesic agents for pain control.”

Reviewer 2 Report
Please, see the attached.

Author Response
Reviewer 2
Comment: This paper discusses " Therapeutic potential of highly selective A3 adenosine receptor 2 ligands in the central and peripheral nervous system ". The main contribution of the paper is "it discusses the effect of last generation A3AR ligands obtained by using different in vitro and in vivo models of various PNS or CNS-related disorders, with particular focus on brain ischemia insults and colitis, where the prototypical A3AR agonist, Cl- 21 IB-MECA, or antagonist, MRS1523, have been used in research studies as reference compounds to explore the effects of latest generation ligands at this receptor. " This is an interesting study and is generally well written and structured. However, in my opinion the paper has some shortcomings in regards to signaling of adenosine receptors and mechanism of these receptors with relation to pain and inflammation. Indeed, short paragraph about adenosine receptors in general and its relation to inflammation, metabolism or signaling in peripheral tissue is recommended to be added. This is important to highlight the significance of adenosine receptors. Moreover, cite more references are recommended. In several instances I also suggested to cite more relevant and recent literature. 1. Adenosine A2B Receptors - Mediated Induction of Interleukin-6 in Skeletal Muscle Cells https://www.ncbi.nlm.nih.gov/pmc/articles/PMC7227993/ 2. The Impact Of Adenosine A2B Receptors Modulation On Nuclear Receptors (NR4A) Gene Expression https://biomedpharmajournal.org/vol9no1/the-impact-of-adenosine-a2b-receptorsmodulation-on-nuclear-receptors-nr4a-gene-expression/ 3. Impact of Adenosine A2 Receptor Ligands on BCL2 Expression in Skeletal Muscle Cells https://www.mdpi.com/2076-3417/11/5/2272 Minor comments:
Our response. We thank the Reviewer for her/his time spent on the paper and constructive comments that helped us to significantly improve our work.
- Please, try to add general paragraph about adenosine receptors including Gs coupled A2A and A2B and discuss it importance to metabolism and inflammation in peripheral tissue such as skeletal muscle since inflammation is related to pain as well. (Above references are suggested)
Our response. We thank the Reviewer for rising this important point. A paragraph titled “Adenosine receptors in peripheral tissues” has been added (paragraph 1.1, page 2 lines 70-83), to discuss the role of Gs coupled A2A and A2B and inflammation in peripheral tissues, as follows:
“1.1 Adenosine receptors in peripheral tissues.
In the periphery, adenosine receptors modulate a number of events including inflammation, metabolism and cell-to-cell signalling. A pervasive action of peripheral adenosine is on A1Rs in the hearth, where they are highly expressed and mediate potent bradycardia. For this reason, adenosine is administered as an emergency drug in conditions of arrhythmia, i.e. paroxysmal supraventricular tachycardia [19].
It is also worth to note that adenosine is one of the most powerful endogenous anti-inflammatory agent thanks to its action on A2ARs [5]. Indeed, A2ARs are highly ex-pressed in inflammatory cells including lymphocytes, granulocytes, and monocytes/macrophages, where they reduce the release of pro-inflammatory cytokines, i.e. tumour necrosis factor-alpha (TNFα), interleukin-1β (IL-1 β), IL-6 [20] and IL-12 [21] and enhance the release of anti-inflammatory mediators, such as IL-10 [22]. Furthermore, A2AR activation on blood vessels is a powerful hypotensive stimulus due to its vasodilating action via intracellular cAMP increase [23].
Concerning the A2BR subtype, the main A2BR-mediated effect in the periphery is known to be in the airways, where this receptor is highly expressed and mediates robust broncocostriction [25]. Indeed, the metilxantin theophylline, a non-selective adenosine receptor antagonist, is a second-line bronchodilator in asthma therapy [26].”
Furthermore, the title “1.2 Adenosine receptors in the CNS” has been added to previous text (paragraph 1.2, page 3 line 101). Above mentioned references have been quoted.
- Please, discuss the role of cAMP in pain and compare A1, A3 (Gi) to A2A and A2B (Gs) to pain
Our response. As suggested, we added a new paragraph about the role of cAMP in pain in relation to the activation of different adenosine receptor subtypes, as follows (page 6 lines 260-289):
“The first proof of adenosine’s involvement in anti-nociception dates from the 1970s, when administration of adenosinergic agonists proved effective in pain control. These studies emphasized the role of adenosine A1Rs in producing anti-nociception with some effects ascribed to the A2AR subtype [82-83]. Adenosine involvement in peripheral nociception was further confirmed, e.g. the exogenous administration of A1R agonists locally to the hind paw of the rat produces anti-nociception in a pressure hyperalgesia model [84] whereas local administration of A2R agonists enhances pain responses [85], an action due to adenosine A2AR activation as confirmed by using the selective agonist CGS21680 [86]. Later on, it was demonstrated that the anti-nociceptive action of A1R agonists has ascribed to AC inhibition and to consequent decrease in cAMP production in sensory nerve terminals [87-88], thus a robust protective role of A1R agonists emerged [89]. On the other hand, the A2AR-mediated promotion of cutaneous pain resulted from stimulation of AC, leading to increased cAMP levels in the sensory nerve terminals [87-88], thus producing opposite effects to those elicited by the anti-hyperalgesic, Gi-coupled, A1R subtype. However, the relation between A2ARs and pain has been controversial, with evidence sustaining either pro-nociceptive or anti-nociceptive activity depending on the receptor localization and animal models of pain [91]. Indeed, a relevant A2AR-mediated anti-nociceptive effect has been described in a recent study demonstrating that central neuropathic pain evoked by dorsal root avulsion could be reversed by a single intrathecal injection of A2AR agonists [93]. The beneficial effects of A2AR agonists in this particular experimental model were associated to reduced reactive gliosis in the CNS. At variance, A2AR antagonists reduced chemotherapy-induced neuropathic pain when administered orally [94]. The discrepancies between the reported effects of A2ARs in pain control could be due to the possible op-posing roles that this adenosine receptor subtype exerts in the periphery (an-ti-inflammatory effect) versus the CNS (pro-excitatory effect) (for reviews, see [95]). On these basis, the idea took shape that AC-stimulating receptors were pro-algesic factors whereas AC-inhibiting messengers would proof as anti-nociceptive signals. In line with this assumption, A2BR activation promotes pain states by increasing the release of inter-leukin-6 (IL-6) [96,97], a pro-inflammatory cytokine also known to cause nociceptor hyperexcitability [96].”
- Please, discuss the role of NR4A in pain and compare A1, A3 (Gi) to A2A and A2B (Gs) to pain.
Our response. As suggested, we added a new paragraph to discuss the above mentioned issue, as follows (pages 6-7 lines 289-299):
“At this regard, it should be noted that adenosine-mediated signalling in inflammation and pain control has been recently linked to the nuclear orphan receptors 4A (NR4A)-dependent pathway [98]. Indeed, it has been demonstrated that A2AR activation counteracts NR4A2 and NR4A3 gene induction in a human mast cell line whereas, after A2AR-silencing or in the presence of the A2AR antagonist SCH58261, the adenosine analogue NECA amplified NR4As induction, thus suggesting that A2AR activation counter-acts NR4A2 and NR4A3 induction in mast cells whereas the activation of other AR sub-types (i.e. A2BRs and/or A3Rs) induces the upregulation of these factors [96]. Such data suggested that the role of adenosine in mast-cell-related inflammatory events could be linked to differential nuclear orphan receptors 4A axis modulation, depending on the adenosine receptor subtype stimulated”.
- Please, discuss the role of inflammation of peripheral tissue such as skeletal muscle and it relation pain. (Short paragraph)
Our response. As suggested, we added the following paragraph (page 7 lines 313-319):
“Of note, mechanical muscle hyperalgesia is a relatively common invalidating condi-tion where the classical inflammatory response (release of bradykinin, prostaglandins, pro-inflammatory cytokines, etc…) plays a major role. At this regard, it has been demon-strated that A2BR stimulation in rat skeletal muscle cells induces IL-6 release by a cAMP-dependent pathway, thus indicating A2BR antagonists as potentially important pharmacological targets in treating inflammation and related diseases in skeletal muscle tissues [24,97].”
- Please, discuss the role of TRPV1 and its relation to pain/A1 (Mechanism of action)
Our response. As suggested, we added the following paragraph (page 7 lines 307-312):
“Another possible pathway of adenosine-mediated pain control has been proposed to consist in direct TRPV1 inhibition by directly interacting with the receptor protein [98]. In particular, it was demonstrated that capsaicin-induced inward currents in DRG neurons were inhibited by adenosine and that A1Rs on DRG neurons co-localize with TRPV1 as a membrane microdomains in allodynic mice, the level of colocalization correlating with the development of the symptom [101].”
- Please, discuss the role of apoptosis (BCL2) to its relation to pain/adenosine receptors, in general.
Our response. As suggested, we added the following paragraph (page 7 lines 299-306):
“Another piece of evidence about the role of adenosine in inflammation and pain came from recent data showing that CCI-induced increase in iNOS, bax/bcl2, iba-1 and TNF-α expression in the lumbar spinal cord of rodents was attenuated by allopurinol, an effect reversed by the A1R-selective and the unselective adenosine receptor antagonist theophylline and by the A1R-selective blocker DPCPX. Of note, the mechanical anti-allodynic effect of allopurinol was only prevented by theophylline, indicating, even in this experimental mode, a differential role of distinct adenosine receptor subtypes [99].”
- In abstract, line 18 The is in bold. Why? Amended.
- Please, add one figure to adenosine receptors in general and inflammation (using peripheral and central tissue). Inflammation is important factor.
Our response. As suggested, we added a new Figure (Figure 3) to describe the role of adenosine receptors in inflammation in peripheral and central tissues. We also added some information about cardiovascular and respiratory systems, to try to make the figure more complete
- Explain why have you chosen A3 in particualr (but not A2…A1,…for example)?
Our response. We thank the Reviewer to highlight this important point. A sentence has been added at page 4 lines 144-153, as follows:
“The fact that A3R agonists proved devoid of significant cardiovascular side effects in clinical trials so far, render this adenosine receptor subtype an eligible target to obtain adeno-sine-mediated protective effects, for example in ischemia or chronic pain, without under-going A1R-mediated bradycardia or hypotension nor A2A-mediated vasodilation [34]. Furthermore, conflicting results obtained with A2R-selective agonists in preclinical models discouraged the attempt to concretize into a clinical approach with these compounds [53].”
- Figure 2 is not obvious to me. It is not understandable.
Our response. We thank the Reviewer for alerting us on this unclear point. As suggested, we modified Figure 2, which is now Figure 3 in the revised manuscript, to make it more understandable.
- The introduction provides a good, generalized background of the topic. However, why not cite more recent literature papers, 2021 (above).
Our response. More recent literature was quoted in the introduction, as follows:
Ref. n. 22: Sohn R, Junker M, Meurer A, Zaucke F, Straub RH, Jenei-Lanzl Z. Anti-Inflammatory Effects of Endogenously Released Adenosine in Synovial Cells of Osteoarthritis and Rheumatoid Arthritis Patients. Int J Mol Sci. 2021 Aug 19;22(16):8956. doi: 10.3390/ijms22168956
Ref. n. 47: Coppi E, Cherchi F, Fusco I, Dettori I, Gaviano L, Magni G, Catarzi D, Colotta V, Varano F, Rossi F, Bernacchioni C, Donati C, Bruni P, Pedata F, Cencetti F and Pugliese AM. Adenosine A2B receptors inhibit K+ currents and cell differentiation in cultured oligodendrocyte precursor cells and modulate sphingosine-1-phosphate signaling pathway. Biochem Pharmacol 2020 Jul 177:113956; doi: 10.1016/j.bcp.2020.113956
Ref. n. 92: Vincenzi, F., Pasquini, S., Borea, P. A., and Varani, K. (2020) Targeting Adenosine Receptors: A Potential Pharmacological Avenue for Acute and Chronic Pain. Int J Mol Sci 2020 Nov 18;21(22):8710. doi: 10.3390/ijms21228710
Ref. n. 97: Kotanska, M., Szafarz, M. L., Mika, K., Dziubina, A., Bednarski, M., Muller, C. E., Sapa, J., and Kiec-Kononowicz, K. (2021) PSB 603-a known selective adenosine A2B receptor antagonist - has anti-inflammatory activity in mice. Biomedicine & Pharmacotherapy 135
Ref. n. 101: Chin-Hong Chang, Ying-Shuang Chang, Yu-Lin Hsieh. Transient receptor potential vanilloid subtype 1 depletion mediates mechanical allodynia through cellular signal alterations in small-fiber neuropathy. Review Pain Rep 2021 Apr 2;6(1):e922. doi: 10.1097/PR9.0000000000000922. eCollection 2021
- I think the motivations for this study need to be made clearer. In particular, the connection between adenosine and pain and inflammation.
Our response. As suggested, we added a number of paragraphs, as per above Reviewer’s comments, that significantly improved our work and also, taken together, clarified the motivations of this study in relation to adenosine, pain and inflammation (please see: page 7 lines 313-319; page 4 lines 144-153; page 7 lines 313-319; page 6 lines 260-289, as reported above).
- Regarding the figures: I recommend make more figures to be illustrative. Given these shortcomings the manuscript requires Minor revisions.
Our response. We thank the Referee for her/his valuable comments on our Figures. As suggested, we added a new Figure in the revised version of the manuscript (now Figure 2) and we modified previous Figure 2 (now Figure 3). We hope we succeeded to make them more illustrative.
"I recommend that this paper be accepted after minor revision."

Reviewer 3 Report
Authors have made a comprehensive review on A3Rs role in brain ischemia and chronic pain.
I consider that the manuscript should be accepted for publication after minor language revision :
Minor language revision
Abstract
-line 18: of is missing
“encouraginf the investigation OF higly selective…”
-line 18: a comma should be deleted in “higly selective agonists and antagonists..”
Main text
-line 115: a dot is missing in “ …[36-37]. As stated above…”
-line 164: “… further evidence shows that…”
-line 164: “…stimulation exacerbateS in vivo…”
-line 167: “….that the blockADE of A3Rs…”
-line 169: a parenthesis is missing after [50]
-line 235: an is should be deleted: In the same work, it was…
-line 245: a comma should be added after ” …neurons [74]..”
-line 251: the sentence seems uncomplete, a derivative of CTX was…..what?
-line 256: an n should be deleted: The authors show that…
-line 270: a reference is missing
-line 270: delete an – in neuroexcitato-ry
-line 320 (fig legend): “which” should be deleted
Author Response
Reviewer 3
I consider that the manuscript should be accepted for publication after minor language revision :
Minor language revision
Abstract
-line 18: of is missing
“encouraging the investigation OF highly selective…”
Our response. We thank the Reviewer for underlining this mistake, which has been amended in the revised version of the manuscript (page 1 line 18).
-line 18: a comma should be deleted in “highly selective agonists and antagonists..”
Our response. We deleted the comma, as suggested (page 1 line 18).
Main text
-line 115: a dot is missing in “ …[36-37]. As stated above…”. Amended (line 160).
-line 164: “… further evidence shows that…”. Amended (line 209).
-line 164: “…stimulation exacerbateS in vivo…”. Amended (line 209).
-line 167: “….that the blockADE of A3Rs…”. Amended (line 212).
-line 169: a parenthesis is missing after [50]. Amended (line 171).
-line 235: an is should be deleted: In the same work, it was… Amended (line 346).
-line 245: a comma should be added after ” …neurons [74]..”. Amended (line 356).
-line 251: the sentence seems uncomplete, a derivative of CTX was…..what?
Our response. We are very sorry for this mistake; a part of the sentence was unintentionally deleted. We added the missing frame, as follows: “a derivative of v-CTX, was FDA approved in 2000 (Prialt) for intrathecal treatment of severe and refractory chronic pain” (page 8 lines 361-363)
-line 256: an n should be deleted: The authors show that… Amended (line 368).
-line 270: a reference is missing. Amended (line 382).
-line 270: delete an – in neuroexcitatory. Amended (line 382).
